# BLADE-ON-PETIOLE proteins act in an E3 ubiquitin ligase complex to regulate PHYTOCHROME INTERACTING FACTOR 4 abundance

**Bo Zhang[1], Mattias Holmlund[1], Severine Lorrain[2], Mikael Norberg[1], László Bakó[3], Christian Fankhauser[2], Ove Nilsson[1]***

[1]Umeå Plant Science Centre, Department of Forest Genetics and Plant Physiology, Swedish University of Agricultural Sciences, Umeå, Sweden; [2]Center for Integrative Genomics, Faculty of Biology and Medicine, University of Lausanne, Lausanne, Switzerland; [3]Umeå Plant Science Centre, Department of Plant Physiology, Umeå University, Umeå, Sweden

**Abstract** Both light and temperature have dramatic effects on plant development. Phytochrome photoreceptors regulate plant responses to the environment in large part by controlling the abundance of PHYTOCHROME INTERACTING FACTOR (PIF) transcription factors. However, the molecular determinants of this essential signaling mechanism still remain largely unknown. Here, we present evidence that the *BLADE-ON-PETIOLE* (*BOP*) genes, which have previously been shown to control leaf and flower development in Arabidopsis, are involved in controlling the abundance of PIF4. Genetic analysis shows that *BOP2* promotes photo-morphogenesis and modulates thermomorphogenesis by suppressing *PIF4* activity, through a reduction in PIF4 protein level. In red-light-grown seedlings PIF4 ubiquitination was reduced in the *bop2* mutant. Moreover, we found that BOP proteins physically interact with both PIF4 and CULLIN3A and that a CULLIN3-BOP2 complex ubiquitinates PIF4 in vitro. This shows that BOP proteins act as substrate adaptors in a CUL3[BOP1/BOP2] E3 ubiquitin ligase complex, targeting PIF4 proteins for ubiquitination and subsequent degradation.
DOI: https://doi.org/10.7554/eLife.26759.001

***For correspondence:**
Ove.Nilsson@slu.se

**Competing interests:** The authors declare that no competing interests exist.

## Introduction

A key element of plant adaptive responses is their ability to make morphological changes by adjusting the regulatory processes controlling their growth and development patterns in response to environmental stimuli such as changes in light and temperature. Members of the phytochrome light receptor family have the unique ability to sense red (R) and far-red (FR) light. They control germination, early seedling development, stem and internode elongation, the balance between leaf lamina and petiole formation (part of the shade avoidance syndrome) and the transition to flowering (*Franklin and Quail, 2010*). All higher plants possess multiple phytochromes (phyA-phyE in Arabidopsis) with phyB being the primary photoreceptor mediating seedling de-etiolation in red light. A major function of phyB is to prevent the shade-avoidance syndrome (SAS) in sunlight, an environment that is rich in red light and thus leading to phyB activation. In Arabidopsis phyB controls the SAS with major contributions from phyD and phyE (*Devlin et al., 1998*). Recently, it was also shown that phyB can integrate light and temperature signals by acting as a thermosensor (*Jung et al., 2016*; *Legris et al., 2016*).

Once activated, phytochromes are transported from the cytosol to the nucleus, where they interact with a group of transcription factors from the Phytocrome Interacting Factor (PIF) family (*Leivar and Monte, 2014*). PIFs act as inhibitors of phytochrome-induced responses in a partially redundant manner and the phytochromes promote these responses by inhibiting the PIFs (*Lorrain et al., 2009*; *Leivar et al., 2008b*; *Leivar et al., 2012*). Different members of the PIF family display different functions in light and temperature regulated development. For instance, PIF1 is a repressor of seed germination (*Oh et al., 2004*), PIF4 plays a crucial role in the response to high temperature (*Koini et al., 2009*) and PIF4 and PIF5 induce leaf senescence in Arabidopsis (*Sakuraba et al., 2014*). PIF4, PIF5 and PIF7 promote elongation of hypocotyls and petioles in response to shade cues (low R:FR and low blue light) (*Lorrain et al., 2008*; *Keller et al., 2011*; *Li et al., 2012*; *de Wit et al., 2016*). PIFs also display different modes of regulation at the transcriptional and post-transcriptional levels. For instance, phosphorylation of all studied PIFs is light regulated and in most cases lead to rapid protein degradation (*Leivar and Monte, 2014*; *Ni et al., 2013*; *Leivar et al., 2008a*). The nature of the protein kinase(s) and the ubiquitin E3 ligase(s) involved in the regulation of PIF stability in response to light have just started to be explored. A recent paper suggests that phytochromes may phosphorylate the PIFs (*Shin et al., 2016*). Also, the BR signaling kinase BRASSINOSTEROID-INSENSITIVE2 (BIN2) has been shown to phosphorylate PIF4 and subsequently affect PIF4 protein abundance, but in a non-light-inducible manner (*Bernardo-García et al., 2014*). As for ubiquitin E3 ligases, the degradation of PIF3 by light was reported to be mediated by the Light Response Bric-a-Brack/Tramtrack/Broad (BTB) proteins (LRBs) interacting with Cullin3 (CUL3), while PIF1 degradation is controlled by a CUL4-COP1-SPA complex (*Ni et al., 2014*; *Zhu et al., 2015*). These findings suggest that the degradation of different PIF proteins might be controlled by specific E3 ligase complexes. However, which E3 ligases that control PIF4 degradation has so far been unknown.

The CUL3-based E3 ligase complexes are composed of a CUL3 backbone, an E2-Ub-docking RING Box1 (RBX1) protein and a member from the large family of BTB-domain containing proteins that serve as target-recognition adaptors (*Genschik et al., 2013*). The Arabidopsis genome contains two *CUL3* genes, called *CUL3A* and *CUL3B* and about 80 genes encoding BTB domain proteins, which could be possible interactors with CUL3A and CUL3B (*Genschik et al., 2013*; *Gingerich et al., 2005*).

Besides LRBs proteins, two other BTB domain containing proteins, NONEXPRESSER OF PR GENES3 and 4 (NPR3 and 4), were shown to function as substrate adaptors in a CUL3 E3 ubiquitin ligase complex to mediate degradation of the co-transcription factor NPR1 (*Fu et al., 2012*). Two close homologs of NPR proteins, BOP1 and BOP2, were previously shown to redundantly regulate leaf development (*Ha et al., 2003*; *Hepworth et al., 2005*; *Norberg et al., 2005*). In Arabidopsis *bop1 bop2* double mutants the leaf lamina extends along the petioles and leaves one and two become massively elongated (*Ha et al., 2003*; *Hepworth et al., 2005*; *Norberg et al., 2005*), phenotypic alterations that are reminiscent of responses to changes in light quality and light intensity (*Franklin and Quail, 2010*). *bop1 bop2* double mutants are also defective in the suppression of bract formation, do not form floral organ abscission zones and display alterations in floral organ identity and positioning (*Hepworth et al., 2005*; *Norberg et al., 2005*; *McKim et al., 2008*). The molecular function of BOP proteins is not known but they have been suggested to act as co-transcription factors (*Khan et al., 2014*) or to inhibit transport of transcription factors to the nucleus (*Shimada et al., 2015*). In this study, we report that BOP proteins act as substrate adaptors in a CUL3 E3 ligase complex to control the degradation of PIF4. This regulatory activity has a strong influence on the role of PIF4 during responses to light and temperature.

## Results

### BOP2 promotes photomorphogenesis in red light

First, we explored the possibility that *BOP* genes may be involved in the regulation of light signaling by analyzing their role in the light-dependent suppression of hypocotyl elongation (*Figure 1a,b*; *Figure 1—figure supplement 1a–c*). Plants were grown in constant monochromatic red, far-red, blue or white light, at a range of fluence rates. In all light qualities, the *bop1* mutant, *bop1-5* (*Norberg et al., 2005*), responded identically to the wild type (Col-0) control (*Figure 1a,b*;

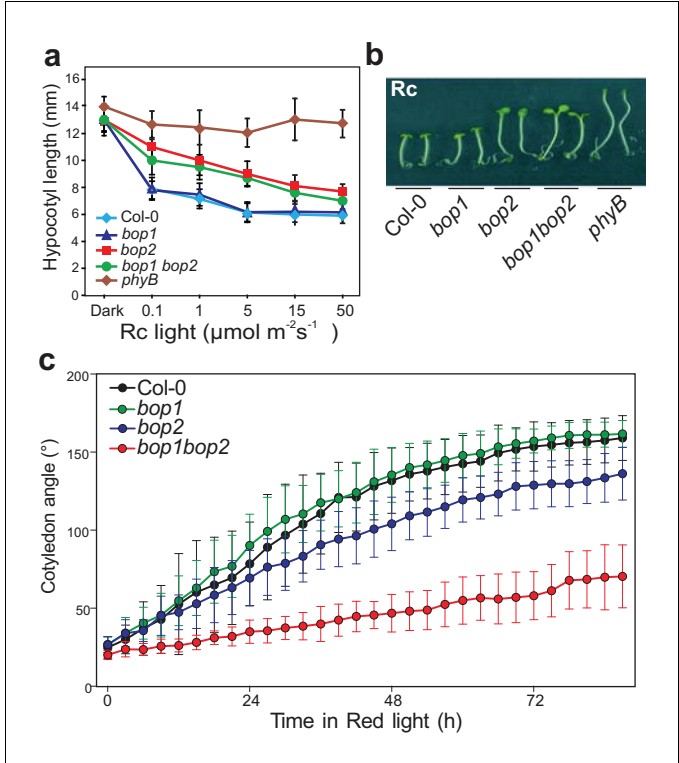

**Figure 1.** BOP2 promotes photomorphogenesis in red light. (a) Hypocotyl lengths of 5-day-old Col-0, bop1, bop2, and bop1 bop2 seedlings grown in constant red light (Rc) of different fluence rates with phyB mutants included as controls. (b) Hypocotyl phenotypes of indicated seedlings in 2 $\mu$mol·m$^{-2}$·s$^{-1}$ red light. (c) Kinematic analysis of cotyledon angles for Col-0, bop1, bop2, and bop1 bop2 plants. Seedlings were firstly grown in dark for 3 days then switched to 6 $\mu$mol·m$^{-2}$·s$^{-1}$ red light. Quantitative data are shown as means ± s.d., n = 25 in (a) and n = 9–14 in (c). The experiments were repeated three times with similar results.

DOI: https://doi.org/10.7554/eLife.26759.002

The following figure supplements are available for figure 1:

**Figure supplement 1.** Hypocotyl lengths and hook angles of BOP mutants in response to white light and monochromatic light.
DOI: https://doi.org/10.7554/eLife.26759.003

**Figure supplement 2.** Hypocotyl lengths, cotyledon angles, and hook angles of BOP1 and BOP2 overexpressing lines in response to red light.
DOI: https://doi.org/10.7554/eLife.26759.004

**Figure supplement 3.** bop2-2 shows wild-type accumulation of phyB levels in 5 day-old seedlings grown in constant red light.
DOI: https://doi.org/10.7554/eLife.26759.005

*Figure 1—figure supplement 1a–c*). In contrast, the bop2 mutant, bop2-2 (*Hepworth et al., 2005*; *Norberg et al., 2005*), showed increased elongation in red light compared to Col-0, while it displayed slightly increased elongation in white light at lower intensity and no hyposensitivity to blue and far-red light (*Figure 1a,b*; *Figure 1—figure supplement 1a–c*). In all these experiments, the bop1 bop2 double mutant behaved identically to the bop2 single mutant, suggesting that only BOP2 plays a significant role in the light-dependent suppression of hypocotyl elongation.

To further investigate the function of the BOP genes in response to red light, we performed kinematic assays for the bop mutants in response to red light (*Boutté et al., 2013*). In these assays, hook unfolding, cotyledon separation, and hypocotyl elongation were measured in a time course. The bop2 and bop1 bop2 mutants displayed significantly decreased cotyledon separation in red light, with a much stronger alteration in the double mutant (*Figure 1c*). The bop1 bop2 double mutant showed a slightly delayed hook opening compared to wild type and the single mutants

(*Figure 1—figure supplement 1d*). These results suggest a partially redundant role of BOP1 and BOP2 in cotyledon separation and hook opening.

Interestingly, plants overexpressing *BOP* genes had the opposite phenotype with both *35S::BOP1* and *35S::BOP2* seedlings displaying reduced hypocotyl elongation at all red light fluence rates tested, and also in darkness (*Figure 1—figure supplement 2a,b*). Kinematic analysis also showed an opposite cotyledon separating phenotype in the *bop1-6D* mutant, a strong activation-tagged line (*Norberg et al., 2005*), with a slightly decreased hook folding in darkness (*Figure 1—figure supplement 2c,d*). The *35S::BOP2* seedlings had a striking hook folding defect in darkness (*Figure 1—figure supplement 2d*), showing that upon over-expression BOP2 can promote photomorphogenesis in the absence of light. Collectively, our data provide genetic evidence that BOP2, and to a lesser extent BOP1, suppress hypocotyl elongation and promotes cotyledon opening, especially in red light conditions.

## BOP2 genetically interacts with PIF4 in response to light

We then characterized the genetic interaction between the red light photoreceptor mutant *phyB* and *bop2*. The *phyB bop2* double mutant showed the same hyposensitive response as the *phyB* single mutant, suggesting that *BOP2* acts in the phyB pathway to suppress hypocotyl elongation in red light (*Figure 1—figure supplement 3a*). In order to test whether the *bop2* phenotype is due to reduced phyB levels we analyzed the levels of the photoreceptor by western blotting. We found that phyB accumulated to normal levels in *bop2* suggesting that BOP2 is rather involved in phyB signaling (*Figure 1—figure supplement 3b,c*).

PIF transcription factors are important mediators of phytochrome signaling and PIF4 plays a prominent role during de-etiolation in red light (*Huq and Quail, 2002*). We therefore investigated the genetic relationship between *bop2* and *pif4*, which show opposite phenotypes in red light (*Figure 2a*) (*Huq and Quail, 2002*). Interestingly, both the *pif4bop2* double mutants and the *pif4-bop1bop2* triple mutants had the same short hypocotyl as the *pif4* single mutant (*Figure 2a,b*; *Figure 2—figure supplement 1*). In the kinematic analysis, compared to wild type the *pif4* mutants showed more separated cotyledons in response to red light (*Figure 2c*). As observed for hypocotyl elongation, the *pif4bop1bop2* triple mutants had the same cotyledon separation phenotype as the *pif4* single mutant (*Figure 2c*). These results show that *pif4* is epistatic over *bop2* and that *PIF4* is necessary for the *BOP2*-mediated suppression of hypocotyl elongation and promotion of cotyledon separation.

## BOP2 promotes red-light induced reduction of PIF4 levels

A possible reason for the epistatic effect of *pif4* over *bop2* is that *bop2* mutants contain increased levels of PIF4. Therefore, we assessed the possibility that the BOPs regulate PIF4 protein accumulation. We generated plants expressing *PIF4-HA* under its endogenous promoter (*PIF4p::PIF4-HA*) in the *pif4* and *pif4bop2* mutant backgrounds. *PIF4p::PIF4-HA* complemented the *pif4* mutant phenotype (*Figure 3a,b*). As previously observed, the level of PIF4-HA rapidly decreased in response to red light (*Figure 3c*) (*Lorrain et al., 2008*; *Nozue et al., 2007*). Interestingly, in the *pif4bop2* mutant background red light led to a slower decline of PIF4-HA levels than in *pif4* (*Figure 3c*). After 10 min of treatment about 30% of PIF4-HA protein remained in the *pif4* background compared to darkness, while more than 80% of PIF4-HA remained in the *pif4 bop2* mutant (*Figure 3d*). These changes in PIF4-HA accumulation in *pif4bop2* were not due to an effect on *PIF4* transcription as *PIF4* transcript levels in the same experimental conditions were the same in both genotypes (*Figure 3e*). In agreement with a role of *BOP2* in the control of PIF4 protein accumulation we also observed an enhanced phenotype of *35S:PIF4-HA* in *bop2* (*Figure 3—figure supplement 1*). Collectively, these results suggest that BOP2 controls PIF4 protein abundance in particular during the transition from dark to light conditions.

## BOP2 physically interacts with PIF4 and CUL3A

PIF4 undergoes proteasome-mediated degradation when etiolated seedlings are transferred into the light (*Lorrain et al., 2009*). Moreover, BOP proteins contain BTB domains, a domain which in other proteins mediates the formation of CUL3-BTB complexes (*Geyer et al., 2003*; *Moon et al., 2004*; *Pintard et al., 2004*). We therefore hypothesized that BOP2 might be part of a CUL3[BOP1/]

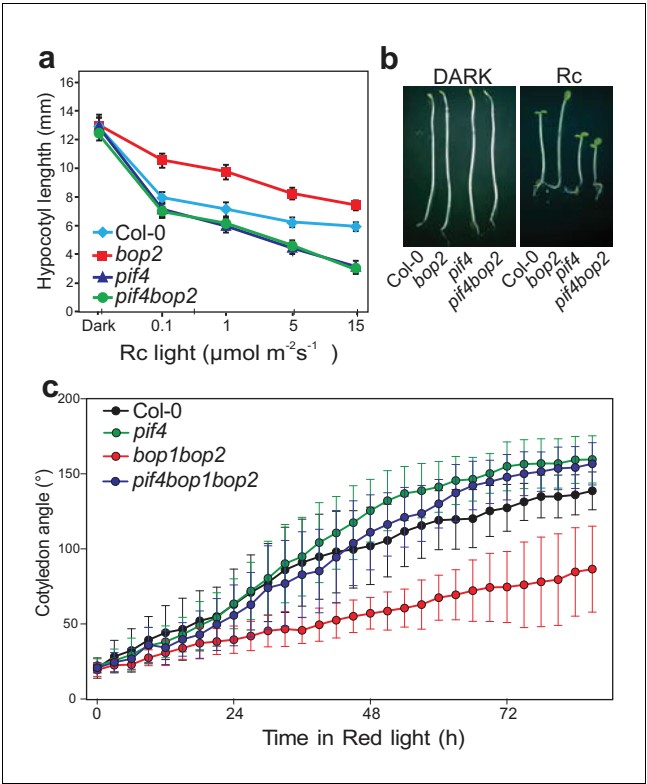

**Figure 2.** BOP2 genetically interacts with PIF4 in response to red light. (**a**) Hypocotyl lengths of 5-day-old Col-0, *bop2*, *pif4*, and *pif4 bop2* seedlings grown in constant red light (Rc) of different fluence rates. (**b**) Hypocotyl phenotypes of indicated seedlings in dark and 2 $\mu$mol·m$^{-2}$·s$^{-1}$ red light. (**c**) Kinematic analysis of cotyledon angles for Col-0, *bop1 bop2*, *pif4*, and *pif4 bop1 bop2* plants. Seedlings were first grown in dark for 3 days then switched to 6 $\mu$mol·m$^{-2}$·s$^{-1}$ red light. Quantitative data are shown as means ± s.d., *n* = 25 in (**a**) and *n* = 9–15 in (**c**). The experiments were repeated three times with similar results.

DOI: https://doi.org/10.7554/eLife.26759.006

The following figure supplement is available for figure 2:

**Figure supplement 1.** Kinematic analysis of hypocotyl lengths.
DOI: https://doi.org/10.7554/eLife.26759.007

BOP2 E3 ubiquitin ligase complex controlling PIF4 degradation. As a first step to test this hypothesis, we used co-immunoprecipitation studies in Arabidopsis protoplasts. HA-CUL3A co-immunoprecipitated together with both myc-BOP1 and myc-BOP2 (*Figure 4a*) and CUL3A could also be co-immunoprecipitated from *35S::myc-BOP2* plants (*Figure 4—figure supplement 1a*) indicating that the proteins indeed interact. In general, CUL3A interacted more strongly with BOP2 than with BOP1 (*Figure 4a*). These findings indicate that BOP1 and BOP2 may act as substrate adaptors in a CUL3$^{BOP1/BOP2}$ E3 ubiquitin ligase complex.

To determine whether the putative CUL3$^{BOP1/BOP2}$ E3 ubiquitin ligase can directly interact with PIF4, we tested if HA-PIF4 could interact with myc-BOP1 or myc-BOP2. HA-PIF4 was found to co-immunoprecipitate with both myc-BOP1 and myc-BOP2 (*Figure 4b*). The interaction between BOP2 and PIF4 was further confirmed in vivo using Bimolecular Fluorescence Complementation (*Figure 4c*; *Figure 4—figure supplement 1b*). A strong signal could be seen in nuclear bodies which were previously observed in PIF4 and phytochrome localization experiments (*Chen, 2008*). In order to test whether PIF4 and BOP2 could interact in the absence of other plant proteins we performed yeast two hybrid assays. The BOP2-PIF4 interaction was detected using both LacZ and histidine auxotrophy as reporters of the interaction (*Figure 4d*). In order to determine whether these proteins directly interact with each other we used in vitro pulldown assays with purified recombinant proteins. This experiment showed that glutathione S-transferase (GST) tagged PIF4 directly interacted with maltose-binding protein (MBP) tagged BOP2, but not with MBP alone (*Figure 4e*).

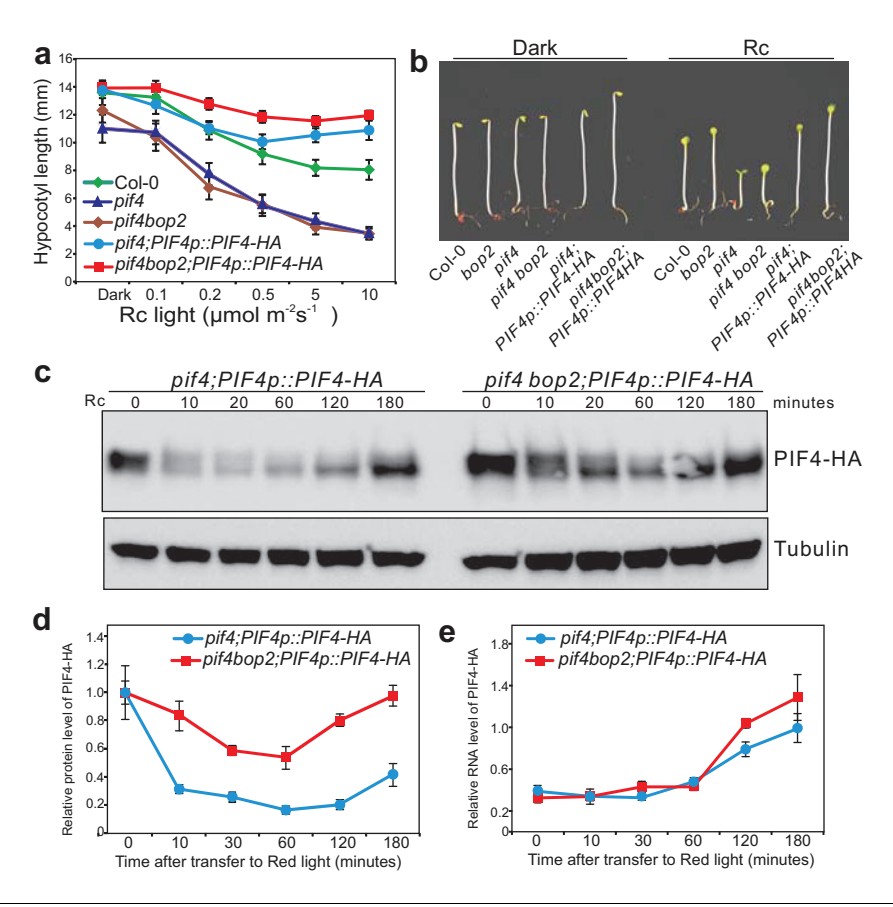

**Figure 3.** BOP2 promotes degradation of PIF4 in response to red light. (**a**) Hypocotyl lengths of 5-day-old Col-0, *pif4*, *pif4 bop2*, *pif4;PIF4p::PIF4-HA*, and *pif4 bop2;PIF4p::PIF4-HA* seedlings grown in constant red light (Rc) of different fluence rates. The data are shown as means ± s.d., *n* = 10. (**b**) Hypocotyl phenotypes of indicated seedlings in dark and 6 μmol·m$^{-2}$·s$^{-1}$ red light. (**c**) Western blot of 3-day-old dark grown seedlings upon red light treatment using an anti-HA antibody for detection of PIF4-HA and anti-tubulin antibody as loading controls. (**d**) Quantification of PIF4-HA protein levels relative to tubulin in 3-day-old dark grown seedlings at indicated time points of 6 μmol·m$^{-2}$·s$^{-1}$ red light treatment. Data were normalized to the Time 0 for each line. (**e**) Quantification of PIF4-HA mRNA levels relative to *PP2A* in 3-day-old dark grown seedlings at indicated time points of 6 μmol·m$^{-2}$·s$^{-1}$ red light treatment. Quantitative data in (**d**) and (**e**) are shown as means ± s.e.m., *n* = 3.
DOI: https://doi.org/10.7554/eLife.26759.008

The following figure supplement is available for figure 3:

**Figure supplement 1.** Hypocotyl lengths of *35S::PIF4-HA* and *35S::PIF4-HA bop2* plants.
DOI: https://doi.org/10.7554/eLife.26759.009

---

Finally, we tested whether BOP2 could mediate the interaction of PIF4 to a CUL3 complex by co-expressing HA-CUL3A, myc-BOP2 and HA-PIF4 in protoplasts followed by immunoprecipitation with anti-myc antibodies (Covance, Princeton, USA). Both HA-CUL3A and HA-PIF4 could be pulled down by BOP2 (*Figure 4f*), suggesting that all three proteins may act in the same complex. Taken together, these results demonstrate that the BOP proteins physically interact with PIF4 and serve as substrate adaptors in a CUL3$^{BOP1/BOP2}$ E3 ubiquitin ligase complex potentially targeting PIF4 for ubiquitination and subsequent degradation.

## A CUL3-BOP complex mediates the polyubiquitination of PIF4

To determine whether BOP2 can direct PIF4 polyubiquitination in vivo, we performed pull-down assays with a Tandem Ubiquitin Binding Entities (TUBEs) approach, to detect polyubiquitinated PIF4-HA proteins in plant expressing *PIF4-HA* under its native promotor. Total ubiquitinated proteins

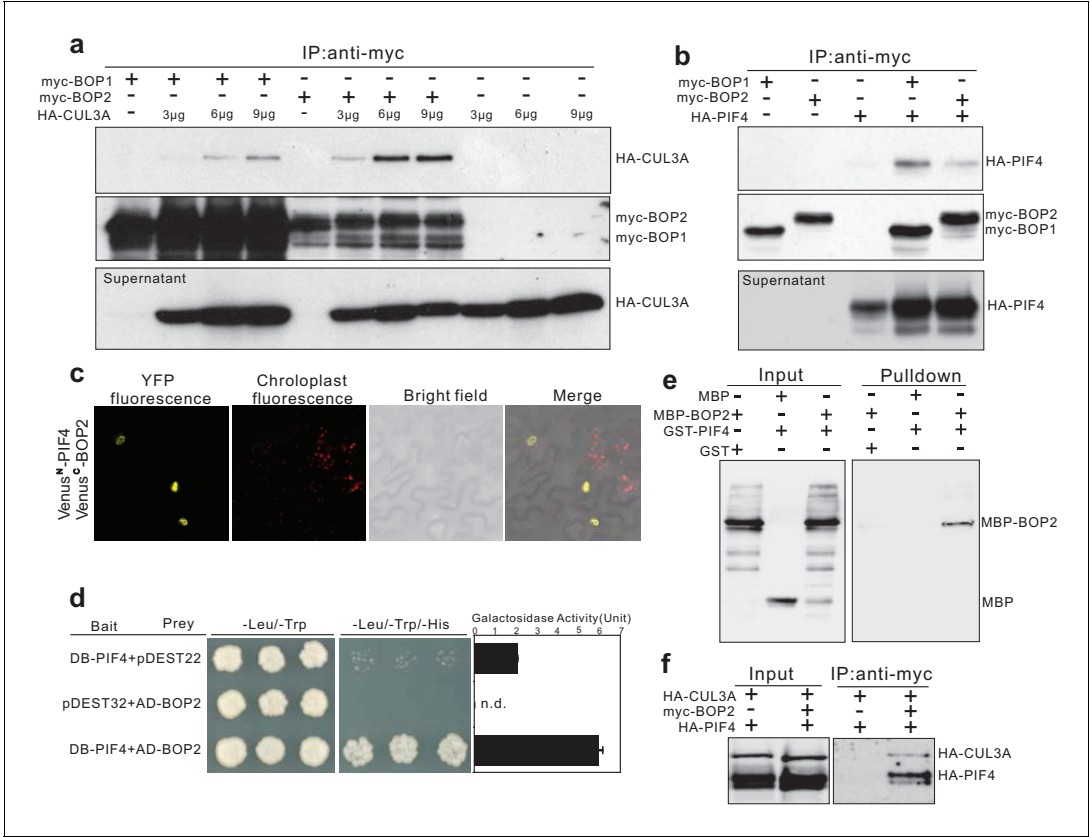

**Figure 4.** BOP1 and BOP2 interact with CUL3 and PIF4. (**a**) Co-immunoprecipitation assays of CUIL3A and BOP1/2 interaction in Arabidopsis protoplasts. Co-immunoprecipitated HA-CUL3A was detected by western blotting using anti-HA antibodies, after immunoprecipitation of myc-BOP1 or myc-BOP2 by anti-myc antibodies. Precipitate probed with anti-myc antibodies and supernatant probed with anti-HA antibodies are shown as controls. 3–9 µg of HA-CUL3A input vector was used. (**b**) Co-immunoprecipitation assays of PIF4 and BOP1/2 interaction in Arabidopsis protoplast as done in (**a**). (**c**) BiFC assays of BOP2 and PIF4 interaction in *Nicotiana benthamiana*. Venus[N], N-terminal part of Venus (aa 1–173); Venus[C], C-terminal part of Venus (aa 156–239). Further controls to show that the interaction is specific are shown in *Figure 4—figure supplement 1b*. (**d**) Yeast two-hybrid assays of BOP2 and PIF4 interaction. The indicated combinations of plasmids were co-transformed into the yeast reporter strain, and interactions between the encoded proteins were assessed by growth on plates with yeast growth media lacking Leu, Trp, and His (-Leu/-Trp/-His). Yeast growth on plates lacking Leu and Trp (-Leu/-Trp) shows the presence of the bait and prey vectors. The strength of activation of the second reporter gene (β-galactosidase) is shown in the chart to the right. Negative controls are represented by the vector combinations containing one of the empty vectors, pDEST22 or pDEST32. Quantitative data are shown as means ± s.e.m, *n* = 3. (**e**) In vitro pulldown assays of BOP2 and PIF4 interaction. MBP-BOP2 or MBP recombinant proteins were pulled down by GST or GST-PIF4 recombinant proteins immobilized on glutathione Sepharose 4B beads, then analyzed by western blotting using anti-MBP antibodies. (**f**) Co-immunoprecipitation assays of CUIL3A, PIF4, and BOP2 interactions in Arabidopsis protoplast. Co-immunoprecipitated HA-CUL3A and HA-PIF4 were simultaneously detected by western blotting using anti-HA antibodies, after immunoprecipitation of myc-BOP2 by anti-myc antibodies.

DOI: https://doi.org/10.7554/eLife.26759.010
The following figure supplement is available for figure 4:

**Figure supplement 1.** Protein-protein interaction analysis.
DOI: https://doi.org/10.7554/eLife.26759.011

from the dark and light-treated seedling extracts were purified using agarose beads coupled with TUBEs (tebu-bio, Le Perray-en-Yvelines, France), then detected by western blotting using anti-HA (Roche, Basel, Switzerland) and anti-ubiquitin antibodies (Santa Cruz biotechnology, Dallas, USA). A set of closely migrating high molecular weight proteins were observed in red light treated seedlings, indicating a light induced polyubiquitination of PIF4-HA (*Figure 5a*, *Figure 5—figure supplement 1*). Significantly reduced polyubiquitination of PIF4-HA was observed in the *bop2* mutant background (*Figure 5a,b Figure 5—source data 1*), indicating that in vivo BOP2 promotes polyubiquitination of PIF4 in response to red light.

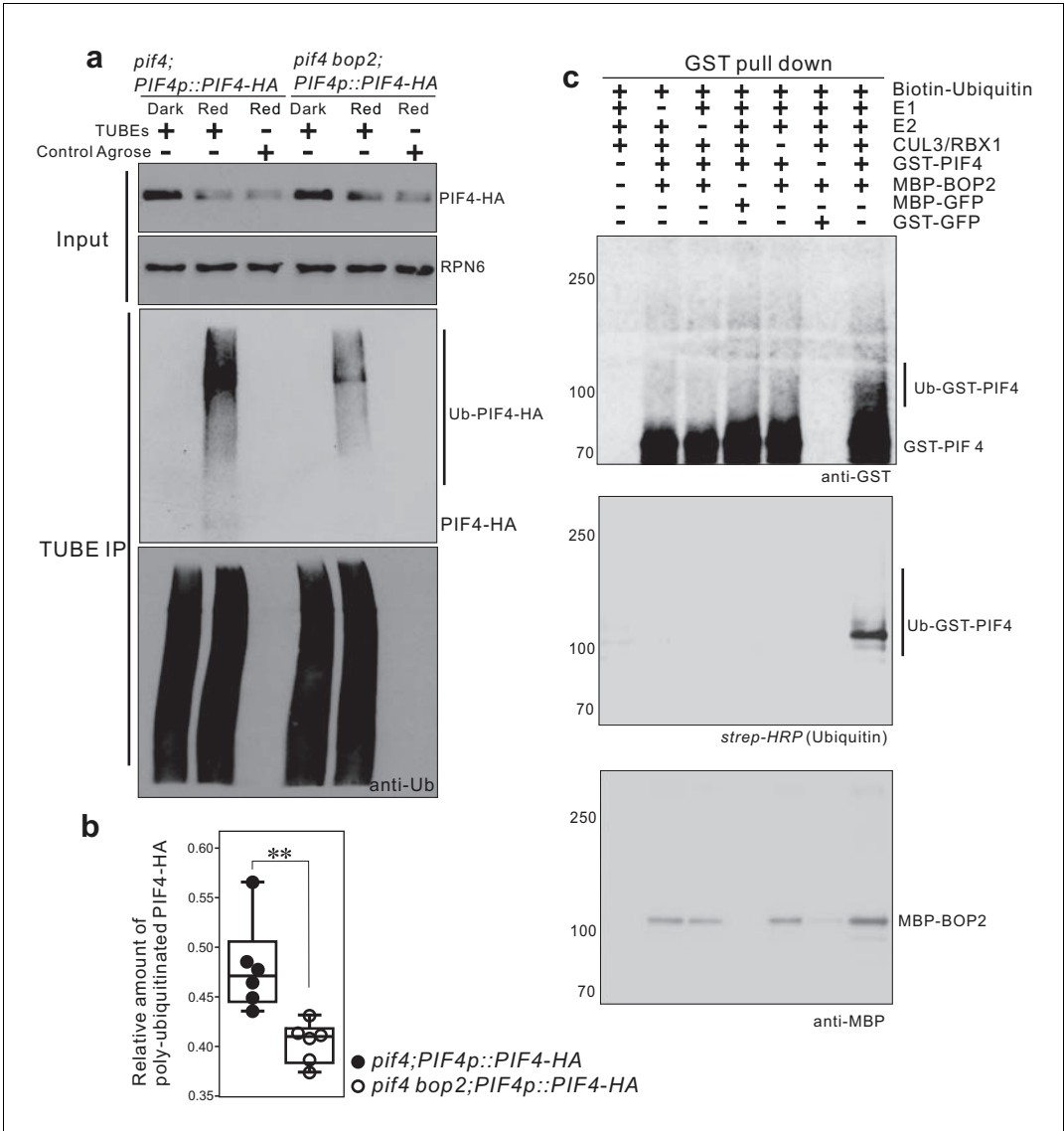

**Figure 5.** CUL3-BOP complex mediates the polyubiquitination of PIF4. (a) TUBEs (tandem ubiquitin binding entities) assays of ubiquitinated proteins in dark and red light treated *pif4;PIF4p::PIF4-HA* and *pif4bop2;PIF4p:: PIF4-HA* seedlings. 3-day-old dark-grown seedlings were irradiated with 6 μmol·m$^{-2}$·s$^{-1}$ red light for 2 min followed by 8 min in the dark before harvesting. Total ubiquitinated proteins from dark and red light treated samples were immunoprecipitated with argrose-TUBE2, then analyzed by western blotting with anti-HA antibodies for detection PIF4-HA and anti-ubiquitin antibodies as loading controls. Control agarose that were not TUBEs conjugated were used as negative controls. Anti-RPN6 antibodies were used as loading controls for input samples. (b) Quantification of ubiquitinated PIF4-HA protein levels relative to total ubiquitinated proteins. Result is shown in a box-and-whiskers plot. Statistical significance was determined using the Student's *t* test (two-sided) between *pif4;PIF4p::PIF4-HA* and *pif4 bop2;PIF4p::PIF4-HA* lines. Circles represent each measured sample from six independent experiments, **p<0.01. (c) In vitro ubiquitiniation assays. A Cullin3 E3 ubiquitin ligase complex was assembled with recombinant MBP-BOP2 and incubated with GST-PIF4. The reactions were then pulled down with Glutathione Sepharose 4B beads followed by Western blotting analysis using anti-GST and anti-MBP antibodies. Streptavidin-HRP were used for detection of biotin-labeled ubiquitinated protein. MBP-GFP and GST-GFP were used as negative controls.

DOI: https://doi.org/10.7554/eLife.26759.012

The following source data and figure supplement are available for figure 5:

**Source data 1.** Quantification of ubiquitinated PIF4-HA protein levels relative to total ubiquitinated proteins from TUBEs assays.

*Figure 5 continued on next page*

*Figure 5 continued*

DOI: https://doi.org/10.7554/eLife.26759.014

**Figure supplement 1.** BOP2 mediates ubiquitination of PIF4-HA also in darkness.

DOI: https://doi.org/10.7554/eLife.26759.013

To investigate whether BOP2 can directly recruit PIF4 for ubiquitination, we then assembled a Cullin3-based E3 ligase in vitro using recombinant MBP-BOP2, GST-PIF4, and human Cullin3/RBX1 proteins (Ubiquigent, Dundee, UK), and performed in vitro ubiquitination assays using biotinylated ubiquitin as substrates (R&D Systems, Abingdon, UK). After incubation with the E3 complex, GST-PIF4 protein was precipitated with glutathione sepharose 4B beads (GE Healthcare, Uppsala, Sweden) and subsequently analyzed by western blotting using anti-GST antibodies (Santa Cruz Biotechnology, Dallas, USA) and horseradish peroxidase (HRP) conjugated streptavidin (Sigma, St. Louis, USA). The result showed the presence of more slowly migrating, ubiquitinated PIF4 isoforms following the in vitro assays (*Figure 5c*). Furthermore, ubiquitinated GST-PIF4 was only observed in the presence of MBP-BOP2. These data demonstrate that BOP2 possesses the capacity to directly ubiquitinate PIF4. Collectively, our results indicate that the CUL3$^{BOP1/BOP2}$ E3 ubiquitin ligase complex controls PIF4 protein levels (*Figures 3–5*).

## BOP2 modulates PIF4 abundance in response to temperature

In addition to its role in light signaling, *PIF4* is known to play a key role in the response to warm temperatures (*Koini et al., 2009*; *Johansson et al., 2014*). We therefore tested whether BOPs modulate this response by controlling PIF4 levels. In constant white light, hypocotyl length of all tested mutants, *pif4*, *bop1*, *bop2*, and *bop1bop2,* was indistinguishable from the wild type at 22°C (*Figure 6a*). However, at 28°C the hypocotyls of *bop1* and *bop2* mutants were longer than the wild type while in *pif4* mutants the temperature response was largely abolished (*Figure 6a*). On the contrary, overexpression of BOP1 or BOP2 strongly suppressed hypocotyl elongation at 28°C (*Figure 6a*). Interestingly, the *bop1bop2* double mutant displayed an enhanced phenotype compared to *bop1* and *bop2* single mutants, indicating that, in contrast to the light response, BOP1 and BOP2 function in a partially redundant manner in response to temperature. Similar results were observed under 12 hr light/12 hr dark growth conditions, except that under these conditions *bop* mutants showed longer hypocotyls also at 22°C (*Figure 6—figure supplement 1a*). This phenotype is likely due to the higher levels of PIF4 accumulated in dark compared to in constant light (*Figure 3c*). Importantly, as observed for the red light responses, the *pif4* phenotype was epistatic over the *bop2* single mutant and the *bop1bop2* double mutant in both growth conditions (*Figure 6a*, *Figure 6—figure supplement 1a*), suggesting that the longer hypocotyls in *bop* mutants is due to elevated PIF4 levels.

We then tested the PIF4-HA protein abundance in *pif4;PIF4p::PIF4-HA* and *pif4bop2;PIF4p::PIF4-HA* lines in constant white light (*Figure 6b,c*; *Figure 6—figure supplement 1b*). As previously observed, the level of PIF4-HA dramatically increased in response to high temperature (*Figure 6b,c*) (*Johansson et al., 2014*; *Oh et al., 2012*). Consistent with the red light response, higher abundance of PIF4-HA was observed in the *pif4 bop2* background compared to the *pif4* background at both 22°C and 28°C (*Figure 6b,c*). We also showed that the longer hypocotyls of *bop1-5* mutants under 12/12 hr growth conditions (*Figure 6—figure supplement 1a*) could be linked to increased levels of PIF4 (*Figure 6—figure supplement 1c,d*) confirming that BOP1 and BOP2 have partially overlapping functions during these conditions. One recent study has shown that high temperature increases the rate of reversion from the active Pfr form to the inactive Pr form of phyB (*Jung et al., 2016*; *Legris et al., 2016*). As PfrB (phyB in the Pfr form) promotes PIF4 degradation (*Leivar et al., 2008b*; *Leivar et al., 2012*), our findings suggest that the longer hypocotyls in *bop* mutants at high temperature result from enhanced PIF4 accumulation due to reduced PfrB and reduced BOP-mediated degradation. All together, these data suggest that BOP-mediated control of PIF4 abundance is important to control not only the light-mediated but also the temperature-mediated growth response in Arabidopsis.

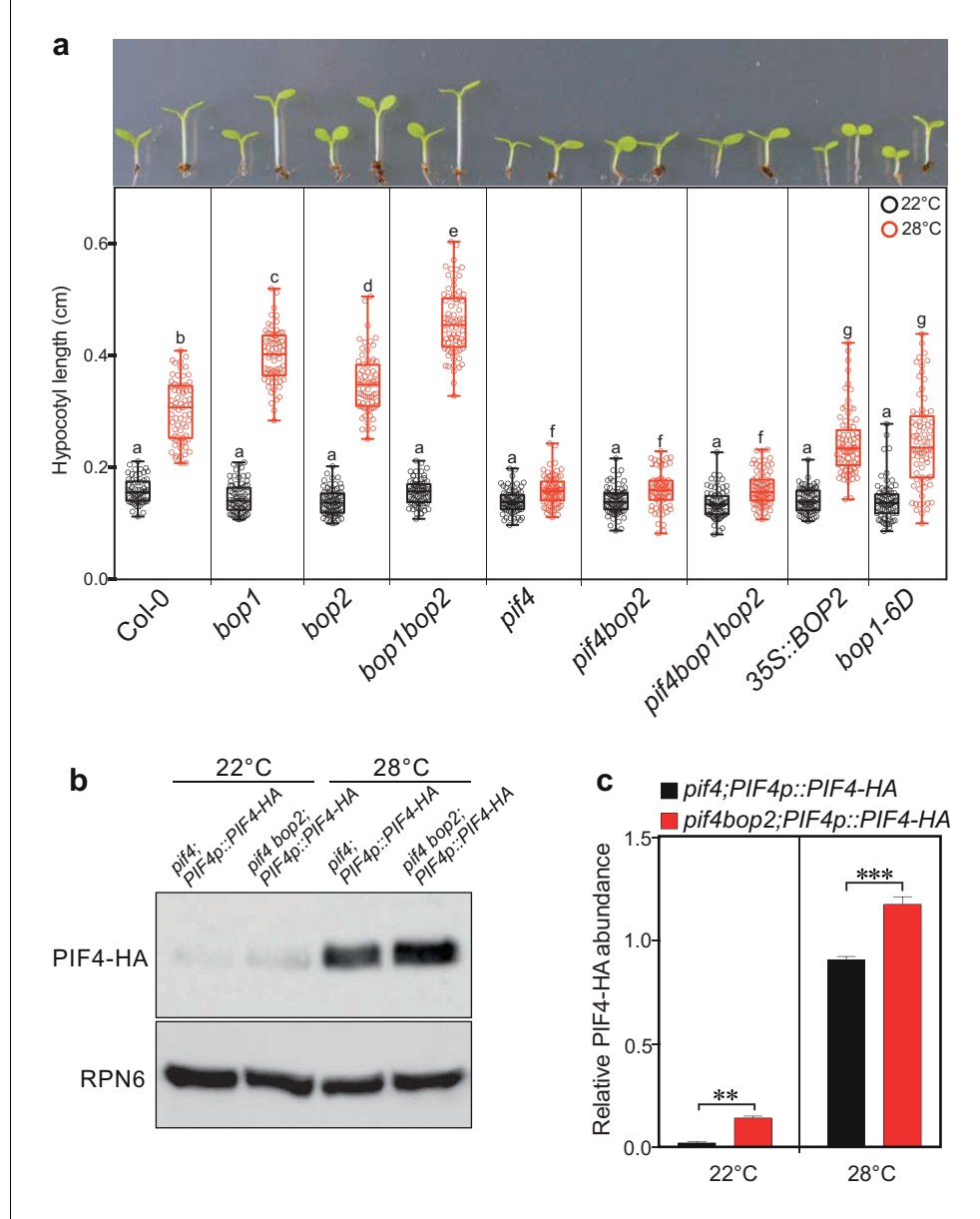

**Figure 6.** BOP2 modulates the PIF4 abundance in temperature response. (**a**) Hypocotyl lengths of indicated seedlings in response to the temperature of 28°C. Seedlings were grown firstly at 22°C in constant white light for 4 days then transferred to 28°C or kept in 22°C for another 4 days. The upper panel shows the hypocotyl length phenotypes of indicated lines. The lower panel shows a box-and-whiskers plot of the data. Multiple comparison was performed in a post ANOVA Fisher's test and lines that do not share any letters are significantly different from each other. Circles represent each measured individual, $n > 60$, $p < 0.01$. (**b**) Western blot of 6-day-old seedlings in response to high temperature using an anti-HA antibody for detection of PIF4-HA and anti-RPN6 antibody as loading control. Seedlings of *pif4;PIF4p::PIF4-HA* and *pif4bop2;PIF4p::PIF4-HA* were firstly grown in constant white light at 22°C for 4 days, then transferred to 28°C or kept at 22°C for another 2 days. (**c**) Quantification of PIF4-HA protein levels relative to RPN6 of the samples from (**b**). Statistical significance was determined using the Student's *t* test (two-sided). Quantitative data in (**d**) and (**e**) are shown as means ± s.e.m. $n = 3$, **$p < 0.01$, ***$p < 0.001$.
DOI: https://doi.org/10.7554/eLife.26759.015

The following figure supplements are available for figure 6:

**Figure supplement 1.** Hypocotyl lengths of indicated seedlings in response to different temperatures.
DOI: https://doi.org/10.7554/eLife.26759.016

**Figure supplement 2.** Genetic interactions between *bop1*, *bop2* and *phyB*.

*Figure 6 continued on next page*

*Figure 6 continued*

DOI: https://doi.org/10.7554/eLife.26759.017

## Discussion

PIF4 is now established as a central component in the regulation of plant growth in response to many different environmental and hormonal signaling pathways, such as the response to light, temperature, gibberellins and brassinolides (*Sakuraba et al., 2014*; *Bernardo-García et al., 2014*; *Johansson et al., 2014*; *Oh et al., 2012*; *Lucyshyn and Wigge, 2009*; *Sun et al., 2013*; *Li et al., 2016*; *Pedmale et al., 2016*). For light signaling it has been known for quite some time that this involves the regulation of PIF4 stability through ubiquitination targeting the protein for degradation (*Lorrain et al., 2009*; *Lorrain et al., 2008*). However, the nature of the E3 ligase complex mediating this ubiquitination has been unknown. Here we show that the BOP proteins can act as substrate adaptors in a CUL3$^{BOP1/BOP2}$ E3 ubiquitin ligase complex that can ubiquitinate and target PIF4 for degradation. This has a significant contribution in regulating the levels of PIF4 particularly during de-etiolation in red light and when growing plants at elevated temperatures (*Figure 3c,d*, *Figure 5a*, *Figure 6b,c*). However, other E3 ligases or proteases are likely to contribute since PIF4 ubiquitination and degradation is not completely abolished in *bop* mutants (*Figure 3c,d*, *Figure 5a*).

In a recent study, the E3 ubiquitin ligase HIGH EXPRESSION OF OSMOTICALLY RESPONSIVE GENES 1 (HOS1) was shown to interact with both phyB and PIF4. However, instead of mediating PIF4's protein degradation, HOS1 was found to suppress transcriptional activity of PIF4 without affecting its protein abundance (*Kim et al., 2017*). In this study, we show that the degradation of PIF4 is mediated by a novel Cullin3-based E3 ubiquitin ligase. All these findings suggest a complexity of the regulation of PIF protein abundance by different E3 ubiquitin ligases. Interestingly, apart from the light-triggered protein degradation, the CUL4-COP1-SPA E3 ligase was suggested to be involved in PIF1 degradation in darkness (*Zhu et al., 2015*). Recent reports also show that DELLA proteins in the GA signaling pathway mediate PIF3 degradation in a non-light-inducible manner (*Li et al., 2016*), and that PIF4 protein abundance can be regulated in a non-light-inducible manner (*Bernardo-García et al., 2014*). In line with this, we observed that PIF4 protein abundance is higher in *bop2* mutant background in darkness (*Figure 3c*) suggesting that the CUL3-BOP E3 ubiquitin ligase might also be involved in the regulation of PIF4 protein stability in the dark.

In previous studies, BOP proteins have been suggested to act as co-transcription factors (*Khan et al., 2014*) and to have a role in directing BZR1 protein translocation from the cytosol to the nuclei (*Shimada et al., 2015*). Our finding that the BOP proteins can act as substrate adaptors in E3 ubiquitin ligase complexes raises the interesting question whether BOP proteins act in the same way in these previously described processes and, if so, what are their targets? (*Hepworth et al., 2005*; *Shimada et al., 2015*; *Xu et al., 2016*).

BOP1/2 belongs to the BTB-ankyrin protein family containing 6 members, including NRP1/2/3/4 (*Khan et al., 2014*). NPR1 was also first identified as a co-transcription factor, acting as a key regulator of systemic acquired resistance (SAR) (*Mou et al., 2003*). Further studies found that NPR1 and its paralogs NPR3/4 are actually the receptors of the immune signal salicylic acid(SA) (*Fu et al., 2012*; *Wu et al., 2012*). Moreover, NPR3/4 were shown to serve as substrate adaptors in a Cullin3 E3 ligase complex targeting NPR1 for degradation (*Fu et al., 2012*). Our findings provide further evidence of E3 ligase activity in this gene family. However, compared to NPR proteins, BOP1/2 play roles in multiple developmental processes.

The BTB-ankyrin family belongs to the BTB domain containing protein super family with about 80 members in Arabidopsis (*Gingerich et al., 2005*). Another subfamily of the BTB proteins, including LIGHT-RESPONSE BTB1, 2, and 3 (LRB1, 2, and 3), were shown to interact with CUL3 and have been suggested to be part of a CUL3 E3 ligase complex with probable targets in the phyB and phyD signaling pathway (*Christians et al., 2012*). A recent study proved that LRB1 and LRB2 can target PIF3 for degradation in a light-dependent manner (*Ni et al., 2014*). The degradation of PIF1 was shown to be mediated by a CUL4-COP1-SPA E3 ubiquitin ligase complex (*Zhu et al., 2015*). Intriguingly, both *spa* and *lrb* mutants are hypersensitive to light, which is somewhat counterintuitive given that they control the degradation of PIFs which promote elongation. One explanation for those phenotypes is that LRBs and SPAs also control the degradation of other targets that play an opposite role

in the control of hypocotyl elongation. Indeed, the LRB-PIF3 interaction induced the degradation of phyB, and SPAs also control the degradation of proteins such as HRF1 and HY5 which promote de-etiolation (*Ni et al., 2014*; *Sheerin et al., 2015*). In contrast, *bop2* mutants are hyposensitive to red light associated with enhanced accumulation of its target PIF4 and show no alterations in phyB levels (*Figure 1—figure supplement 3b,c*) suggesting that BOP2 plays a less pleiotropic role in the control of de-etiolation than LRBs and SPAs.

It is common among substrate adaptors in Cullin E3 ligase complexes to form both homo- and hetero-dimers, something that can potentially increase their target range (*Bosu and Kipreos, 2008*; *Hua and Vierstra, 2011*). One can speculate, as has been done for other BTB substrate adaptors (*Ni et al., 2014*; *Christians et al., 2012*), that BOPs sometimes work as homodimers, which would explain the unique role of BOP2 in controlling red light suppression of hypocotyl elongation (*Figure 1*), while sometimes BOP1 and BOP2 can work as heterodimers to control other environmental response, which would explain the role of both BOP1 and BOP2 in controlling high temperature induction of hypocotyl elongation (*Figure 6*).

Collectively, these findings suggest that E3 ligases with different substrate adaptors target proteins for degradation at different steps in the phytochrome/PIF signaling pathway. This working model is also confirmed from our studies of phyB/BOP interactions during later stages of development. When grown in constant red light *phyB* mutants displayed a constitutive shade-avoidance response with long petioles but maintained a normal rosette habit with no internodal elongation (*Figure 6—figure supplement 2*). In contrast, dramatic elongation of rosette internodes was observed in all combinations of *phyB* and *bop* mutants (*Figure 6—figure supplement 2*). The *phyB bop* mutant combinations appeared similar to *phyB phyD phyE* triple mutants, which also show rosette internodal elongation (*Devlin et al., 1998*). This suggests that *BOP1* and *BOP2* are involved in phyB/D/E-mediated suppression of the shade avoidance syndrome in red light and, in contrast to the regulation of hypocotyl elongation, *BOP1* and *BOP2* are needed together in this suppression. This suggests that the BOP proteins can also affect other phytochrome signal transduction pathways, besides the phyB pathway.

One outstanding question relates to the role of phosphorylation in the BOP-directed PIF4 ubiquitination and degradation. Our data indicate that PIF4 phosphorylation is not a pre-requisite for BOP2-mediated PIF4 degradation (*Figure 4e*, *Figure 5c*). Interestingly, we also observed a very low level of PIF4-HA polyubiquitination in dark, which was clearly reduced in the *pif4 bop2* background compared to the *pif4* background (*Figure 5—figure supplement 1*). These data indicate that although BOP-mediated PIF4 degradation is stronger in the light, it also occurs in darkness. However, the physiological consequences of this regulatory mechanism are particularly strong during de-etiolation and growth at elevated temperatures. It has been shown that protein phosphorylation is absolutely required for the interaction of PIF3 with its E3 ligase LRB2 (20). Phosphorylation also affects the stability of PIF4 during brassinolide signaling (*Bernardo-García et al., 2014*), although it is unclear if this is absolutely required for PIF4 degradation. Our in vitro results using recombinant (unphosphorylated) *E. coli*-produced proteins suggest that phosphorylation is not required for BOP2 binding to, and ubiquitination of, PIF4. This is then in contrast to the situation with PIF3. This does not exclude that phosphorylation will further enhance BOP binding to PIF4 in vivo, or that it has an effect on ubiquitination and degradation following BOP binding. This will be an important question for future research. Our results show that BOP controls PIF4 stability in both light- and temperature responses suggesting that BOP might serve as a more general regulator of PIF4 accumulation, also affecting other pathways acting through PIF4.

## Materials and methods

### Plant material and growth conditions

The mutants and transgenic plants used in this study have been previously described, and were: *bop1-5* (*Norberg et al., 2005*), *bop1-6D* (*Norberg et al., 2005*), *bop2-2* (*Hepworth et al., 2005*, *Norberg et al., 2005*), *35S::BOP1* (*Norberg et al., 2005*), *35S::BOP2* (*Norberg et al., 2005*), *pif4* (*Lorrain et al., 2008*), *35S::PIF4-HA* (*Lorrain et al., 2008*), *cry1* (*Mockler et al., 1999*), *cry2* (*Mockler et al., 1999*), *phyA* (Salk line N520360), *phyB* (Salk line N569700), the two Salk lines were obtained from NASC (*Supplementary file 1*). The other independent *PIF4-HA* transgenic line (*pif4;*

*PIF4p::PIF4-HA*) was generated by introducing the construct *PIF4p::PIF4-HA* into the *pif4-101* background (*Huang et al., 2016*). The *PIF4p::PIF4-HA* plasmid was constructed using pPZP211 as binary vector. It includes 2.1 kb of *PIF4* promoter sequence, the *PIF4* cDNA, a C-terminal triple *HA* and an *RBCS* terminator sequence. This was all assembled with conventional cloning introducing RE sites that were introduced by PCR. The final construct was sequenced. This line was then crossed to the *bop2-2* mutant. As wild type controls Columbia (Col-0) plants were used.

Plants were grown in 16 hr light/8 hr dark, 12 hr light/12 hr dark, or 8 hr light/16 hr dark cycles on either soil mixed with vermiculite (3:1) or on ½ MS 0.8% agar plates without sugars. Plants grown on ½ MS were surface-sterilised and stratified for 5 days at 4°C in darkness then subjected to 1 hr of white light to induce germination. After a further 23 hr in darkness at 22°C they were placed in constant white light or monochromatic red, blue or far red light of different fluence rates. The light intensities were measured with a spectroradiometer. Only the five longest 5-day-old hypocotyls of each genotype of seedling in each experiment were measured of a total of 25 seedlings to minimize germination effects. The experiments were then repeated five times to give a total of 25 measurements. Kinematic assays were done as previously described with a small modification (*Boutté et al., 2013*). Seedlings were grown in dark for 72 hr then transferred to 6 $\mu$mol·m$^{-2}$·s$^{-1}$ red light. Photos were taken every 3 hr. Hypocotyl lengths, hook angles, and cotyledon angles were measured using ImgaeJ software. Statistical analysis and blot-whisker plots were done using the GraphPad Prism software.

## PIF4-HA stability assay

For the protein stability assays in response to red light, three-day-old *pif4;PIF4p::PIF4-HA*, and *pif4 bop2;PIF4p::PIF4-HA* seedlings were grown in dark on ½ MS medium without sugar and then subjected to 10 $\mu$mol m$^{-2}$ s$^{-1}$ of red light at 22°C. The seedlings were harvested in a time course after subjected to red light. For the PIF4 protein stability assays between Col-0 and *bop1-5* plants, 8-day-old soil-grown seedlings in 12 hr light/12 hr dark condition at 22°C were harvested. Proteins were extracted from 20 seedlings for each line in a PBS buffer containing 0.1% w/v SDS, 0.1% v/v Triton X-100, 1 mM phenylmethylsulfonyl fluoride (Sigma, St. Louis, USA), 14 mM 2-mercaptoethanol, and 2x complete protease cocktail (Roche, Basel, Switzerland). Each extract was cleared by centrifugation at 4°C with full speed for 10 min. Protein concentration was measured in a spectrophotometer with Lowry dye reagent (Bio-Rad, Hercules, USA). Around 30 $\mu$g of total protein was loaded on an 8% SDS-PAGE gel and blotted onto an Immobilon-P PVDF transfer membrane (Millipore, Billerica, USA). The resulting immunoblot was probed with the 16B12 anti-HA-POD monoclonal antibody (Roche, Basel, Switzerland) for detection of PIF4-HA protein or anti-PIF4 antibody from goat (Agrisera) for detection of PIF4 native protein. Band signals were visualized by the SuperSignal Western Blotting system (Thermo Scientific, Waltham, USA). The intensities of Western blot band signals were collected from the LAS-3000 Imaging System (Fuji, Minato, Japan) and were measured using Image J. Quantification was performed with three biological replicates for each line using anti-tubulin (T5168, Sigma, St. Louis, USA) or anti-RPN6 (26S proteasome non-ATPase regulatory subunit) antibodies (BML-PW8370, ENZO Life Sciences, New York, USA) as loading controls.

## PhyB stability assay

40 seeds per genotype/point were plated in 1/2 MS and cold-treated for 3 days in the dark. Germination was induced by 3 hr of 50 $\mu$mol.m$^{-2}$s$^{-1}$ red light. After this time, plates were placed into different light conditions (red 0.1 or 15 $\mu$mol.m$^{-2}$.s$^{-1}$) for 5 days. Seedlings were ground in liquid nitrogen (quiagen tissulyser) and resuspended into 150 $\mu$L of hot 2x FSB buffer. 15 $\mu$L of each sample was separated by SDS-PAGE (10% gel). Quantification was performed as described in (*Trupkin et al., 2007*) using DET3 as a loading control.

## Protoplast transfection and co-immunoprecipitation

*Arabidopsis thaliana* ecotype Columbia cell suspension cultures were used for protoplast isolation. Isolation and transfection of Arabidopsis protoplasts was performed essentially as described previously (*Cruz-Ramírez et al., 2012*). In brief, $5 \times 10^5$ cells were transfected with 3 $\mu$g each of *myc-BOP1*, *myc-BOP2* and *HA-PIF4* or 3–9 $\mu$g of *HA-CUL3A* expression constructs. Transfected cells were cultured for 16 hr at RT, then collected by centrifugation and lysed in 50 $\mu$l of extraction buffer

(EB) containing 25 mM Tris-HCl pH 7.8, 10 mM MgCl$_2$, 5 mM EGTA, 75 mM NaCl, 60 mM beta-glyc-erophosphate, 2 mM DTT, 0.2% Igepal CA 630, 0.1 mM Na$_3$VO$_4$, 1 mM benzamidine and protease inhibitors (Protease Inhibitor Coctail for plant cells, Sigma). For in vivo BOP2-CUL3A interaction analysis, 4-day-old dark-grown seedlings were harvested from Col-0 and *35S::myc-BOP2* transgenic plants. For co-immunoprecipitation assays, 50–75 µg of total proteins were incubated for 2 hr at 4°C with 1.5 µg of antibody against c-Myc epitope (Covance, clone 9E10) in a total volume of 100 µl of EB buffer supplemented with 150 mM NaCl and 0.2 mg ml$^{-1}$ BSA. Immunocomplexes were then adsorbed on 10 µl of Protein G-Sepharose matrix (GE Healthcare), washed three times with Tris buffered saline containing 50 mM Tris-HCl pH 8.0, 150 mM NaCl, 5% glycerol, 0.1% Igepal CA-630 and eluted by boiling in 30 µl of 1.5x Laemmli sample buffer. Proteins were then resolved by SDS-PAGE and blotted to PVDF transfer membrane (Immobilon-P, Millipore). The epitope-tagged proteins were probed with anti-HA-peroxidase (3F10, Roche) or chicken anti-c-Myc antibodies (A2128, Invitrogen, Carlsbad, USA) and detected with the SuperSignal West Pico Chemiluminescent Substrate. The native Cul3A proteins were probed with an anti-AtCul3 polyclonal antibody (Enzo Life Science). To assess abundance of tagged proteins in the supernatants of immunoprecipitation reactions, proteins were precipitated with 10% trichloroacetic acid, resolved by SDS-PAGE and immunoblotted with anti-HA-peroxidase antibody.

## Bimolecular fluorescence complementation assay

The full-length coding sequence (CDS) of *AtBOP2* and *AtPIF4* were amplified and cloned into pDONR207 vector by BP Clonase II (Invitrogen) to construct entry clones pENTR207AtBOP2 and pENTR207AtPIF4. Then pENTR207AtBOP2 and pENTR207AtPIF4 were recombined into pDEST-VYCE(R)$^{GW}$ and pDEST-VYCE(R)$^{GW}$ destination vectors, respectively (*Gehl et al., 2009*). The binary vectors expressing the fusion genes, *Venus$^C$-BOP2* and *Venus$^N$-PIF4*, were transferred into *Agrobacterium tumefaciens* strain GV3101 (pMP90). The constructs, expressing the fusion genes, *Venus$^N$-CNX6* and *Venus$^C$-CNX6*, were transferred into the same strain and used as controls. Then, the fusion genes were co-transfected in different combinations into 4-week-old *Nicotiana benthamiana* leaves by agroinfiltration as previously described (*Gehl et al., 2009*). Fluorescence of the lower epidermis of leaf discs 2 days after infiltration was visualized with a TCS SP2 confocal microscope (Leica, Wetzlar, Germany) and a 488 nm Ar/Kr laser line. Venus$^{N/C}$ fluorescence was detected with the excitation/emission combination, 514/525–535. The chlorophyll auto-fluorescence was detected with the emission, 685–700.

## Yeast two-hybrid analysis

The full-length CDS of *AtBOP2* and *AtPIF4* were cloned into pDEST32 and pDEST22 (Invitrogen, Carlsbad, USA) using Gateway system to generate the GAL4-DB-BOP2 bait and GAL4-AD-PIF4 prey construct, respectively. Then both constructs were co-transformed into the MaV203 yeast strain that contains single copies of each of three reporter genes (*HIS3*, *URA3* and *lacZ*). The yeast cell harboring two constructs were grown on leucine and tryptophan dropout media for transformants selection, and leucine, tryptophan, histidine dropout media for interaction selection. β-galactosidase quantitative assays were performed as described in the Clontech Yeast Protocols Handbook (http://www.clontech.com/SE/Support/ProductDocuments?sitex=10023:22372:US). The combinations with two empty vectors, pDEST32 and pDEST22, were used as negative controls.

## In vitro pulldown assay

His-GST-PIF4 and His-MBP-BOP2 protein were generated from *Escherichia coli* BL21 cells and purified using Ni-NTA Agarose (Qiagen, Hilden, Germany) according to the manufacturer's protocol. 1 µg of GST (Santa Cruz Biotechnology, Dallas, USA) or GST-PIF4 was firstly incubate with 10 µl Glutathione Sepharose 4B beads for 1 hr at 4°C in PBS containing 0.5% Triton X-100. Then the beads were blocked for 30 min in the PBS buffer containing 1% milk powder and 1% Triton X-100. After 5 min' centrifugation with 500 g, the supernatant was discarded. The beads were then washed two times with PBS buffer containing 1% Triton X-100 and incubated with the PBS buffer containing 2% BSA and 1% Triton X-100 for another 30 min. After wash with PBS buffer containing 1% Triton X-100, 1 µg of MBP (New England Biolabs, Ipswich, USA) or MBP-BOP2 were added to the beads and incubated in 300 µl binding buffer (50 mM Tris-Cl, pH 7.6 100 mM NaCl, 1 mM EDTA, 1% Triton

X-100, 0.5 mM DTT, 5% Glycerol) at 4°C for 2 hr. After three times washing with the binding buffer, proteins were eluted with 2x laemli loading buffer then subjected to Western blot analysis with anti-MBP antibodies (New England Biolabs, Ipswich, USA).

## TUBEs analysis

The Immunoprecipitateion of ubiquitinated proteins from *pif4;PIF4p::PIF4-HA* and *pif4 bop2;PIF4p::PIF4-HA* seedlings using Tandem Ubiquitin Binding Entities (TUBEs) agarose (tebu-bio, Le Perray-en-Yvelines, France) were performed as previously described with slight modification (*Ni et al., 2014*). 3-day-old dark-grown seedlings were irradiated with 6 μmol·m$^{-2}$·s$^{-1}$ red light for 2 min followed by 8 min in the dark before harvesting. Proteins were extracted with a buffer containing 100 mM MOPS, pH7.6, 150 mM NaCl, 0.1% NP40, 1% Triton X-100, 0.1% SDS, 20 mM Iodoacetamide, 1 mM PMSF, 2 μg/l aprotinin, 40 μM MG132, 5 μM PR-619, 1 mM 1,10-Phenanthroline, and 2X Complete protease inhibitor Cocktail and PhosStop cocktail (Roche). 30 μl Agarose-TUBE2 was incubated with 2 mg total protein from each sample for 6 hr at 4°C. The agarose beads were washed with extraction buffer four times and eluted with 2x laemli loading buffer then subjected to Western blot analysis with the 16B12 anti-HA-POD antibodies (Roche) for detection of PIF4-HA and anti-ubiquitin antibodies (sc-8017, Santa Cruz Biotechnology) for detection of ubiquitinatied protein. The intensities of western blot band signals collected from the LAS-3000 Imaging System (Fuji) and were measured using Image J. Quantification was performed with the measurements of 6 independent experiments using anti-ubiquitin antibodies as loading controls. Statistical analysis and blot-whisker plots were done using the GraphPad Prism software.

## In vitro ubiquitination assays

His-GST-PIF4, His-GST-GFP, His-MBP-BOP2, and His-MBP-GFP were generated from *Escherichia coli* BL21 cells and purified using Ni-NTA Agarose from Qiagen according to the manufacturer's protocol. Human Cullin3/Rbx1 recombinant proteins were purchased from Ubiquigent (UK). The Neddy-lation of the Cullin3 was performed as previously described using a NEDD8 Conjugation Reaction Buffer Kit from R&D Systems Europe (Abingdon, UK) (*Duda et al., 2008*). The recombinant human ubiquitin-activating enzymes (UBE1) and ubiquitin-conjugating enzymes (UbcH5b) were purchased from R&D Systems Europe. Ubiquitination reactions were performed as described previously with slight modification (*Ni et al., 2014*). About 100 nM UBE1, 1 μM UbcH5b, 100 nM Cul3/Rbx1, 290 nM GST-PIF4, 500 nM MBP-BOP2, were incubated at 30°C for 1 hr in a buffer containing 40 μM biotin-ubiquitin, 5 mM Mg-ATP, 50 mM Tris-HCl pH7.6, 200 mM NaCl, 10 mM MgCl2, 1 Unit Inorganic pyrophophatase, and 1 mM DTT. Reactions were then pulled down with 10 μl Glutathione Sepharose 4B beads. GST-PIF4 and ubiquitinated proteins were detected by Western blot analysis using anti-GST antibodies and streptavidin-HRP conjugates, respectively. Anti-MBP antibodies were used for detection of the MBP proteins after pulldown. The reactions without E1, E2, or Cul3/Rbx1, and the reactions with MBP-GFP or GST-GFP were used as negative controls.

## Acknowledgements

This work was supported by grants from the Swedish Research Council, The Knut and Alice Wallenberg Foundation, and the Swedish Governmental Agency for Innovation Systems to ON and from the Swiss National Science Foundation (grant 310030B141181) to CF. We thank Andrea Maran (University of Lausanne) for his contribution in the initial phase of the project, Akira Nagatani (University of Kyoto) for providing the phyB antibody and Karin Schumacher (University of Heidelberg) for providing the DET3 antibody.

## Additional information

### Funding

| Funder | Author |
| --- | --- |
| Vetenskapsrådet | Ove Nilsson |

| Knut och Alice Wallenbergs Stiftelse | Ove Nilsson |
|---|---|
| VINNOVA | Ove Nilsson |
| Schweizerischer Nationalfonds zur Förderung der Wissenschaftlichen Forschung | Christian Fankhauser |

The funders had no role in study design, data collection and interpretation, or the decision to submit the work for publication.

## Author contributions

Bo Zhang, Data curation, Formal analysis, Validation, Investigation, Visualization, Methodology, Writing—original draft, Writing—review and editing; Mattias Holmlund, Severine Lorrain, Mikael Norberg, Formal analysis, Validation, Investigation, Visualization, Writing—original draft; László Bakó, Resources, Formal analysis, Supervision, Validation, Investigation, Visualization, Methodology, Writing—original draft; Christian Fankhauser, Conceptualization, Resources, Formal analysis, Supervision, Funding acquisition, Validation, Investigation, Writing—original draft, Project administration, Writing—review and editing; Ove Nilsson, Conceptualization, Resources, Data curation, Formal analysis, Supervision, Funding acquisition, Validation, Investigation, Methodology, Writing—original draft, Project administration, Writing—review and editing

## Author ORCIDs

Christian Fankhauser https://orcid.org/0000-0003-4719-5901
Ove Nilsson http://orcid.org/0000-0002-1033-1909

## Decision letter and Author response

Decision letter https://doi.org/10.7554/eLife.26759.020
Author response https://doi.org/10.7554/eLife.26759.021

# Additional files

## Supplementary files

• Supplementary file 1. All primer sequences used in this study.
DOI: https://doi.org/10.7554/eLife.26759.018

• Transparent reporting form
DOI: https://doi.org/10.7554/eLife.26759.019

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
