## [Decision Letter]

Thank you for submitting your article "Blade-on-Petioleproteins act in an E3 ubiquitin ligase complex to regulate Phytochrome Interacting Factor4 Abundance" for consideration by *eLife*. Your article has been reviewed by Christian Hardtke (Senior Editor) and two reviewers, one of whom, Hao Yu (Reviewer #1), is a member of our Board of Reviewing Editors. The following individual involved in review of your submission has agreed to reveal his identity: On Sun Lau (Reviewer #2).

The reviewers have discussed the reviews with one another and the Reviewing Editor has drafted this decision to help you prepare a revised submission.

Summary:

PIF4 is a crucial regulator in light signal transduction, and is gaining more interests recently because of its involvement in high temperature signaling. In this manuscript, Zhang et al. identifies the BOP proteins as CUL3-based substrate receptors that target PIF4 for the ubiquitin-proteasome mediated degradation. They first showed that BOP2 is important in red light-mediated photomorphogenic responses and that pif4 is epistatic to bop2. Then, they demonstrated that the red light-induced reduction of PIF4 proteins are dampened in bop2. Through a series of in vitro and in vivo interaction assays, they further showed that BOPs can interact with CUL3A and PIF4, that the level of ubiquitinated PIF4 is reduced in bop2 and that PIF4 can be ubiquitinated in the presence of BOP2 and the CUL3 machinery. Finally, they reported that in contrast to pif4, bop mutants are hypersensitive to heat and bop2 has a higher level of PIF4 proteins.

Taken together, this manuscript is well-written with a clear logic and provides important insights into regulation of PIF4 abundance in response to light and temperature. However, based on the previous findings on various E3 ligase complexes for other PIFs, justification of this manuscript to be published in *eLife* should be strengthened by more convincing and solid data as stated below.

Essential revisions:

1) One important question that the authors have not addressed is the effect of phosphorylation of PIF4 on the proposed recognition and degradation. It is well-established that light induces rapid phosphorylation and degradation of PIFs, including PIF4, and the more recent work on PIF3 demonstrated the importance of PIF3 phosphorylation in mediating the light-dependent interaction with its substrate receptors (Ni et al., Science 2014). Thus, it is important to test how PIF4 phosphorylation affects its binding to BOPs. This shall help reveal the functional role of the BOP-mediated degradation and potentially address how the degradation is controlled by light.

2) Based on the phenotypes of bop1, bop2 and their double mutants, BOP1 and BOP2 may preferentially mediate PIF4 abundance under different environmental conditions. This manuscript at this stage does not present consistent biochemical data along with the genetic data. Almost all biochemical data are focused on BOP2, while genetic data usually include both bop1 and bop2. As the authors have by default included BOP1 and BOP2 in the E3 ubiquitin ligase complex, they should present evidence that BOP1 also affects PIF4 protein abundance under certain conditions.

3) The authors should preferably provide co-IP data for the interaction between BOP1/2 with PIF4 and CUL3A using protein extracts from transgenic lines rather than *Arabidopsis* protoplasts.

---

## [Author Response]

*Essential revisions:*

*1) One important question that the authors have not addressed is the effect of phosphorylation of PIF4 on the proposed recognition and degradation. It is well-established that light induces rapid phosphorylation and degradation of PIFs, including PIF4, and the more recent work on PIF3 demonstrated the importance of PIF3 phosphorylation in mediating the light-dependent interaction with its substrate receptors (Ni et al., Science 2014). Thus, it is important to test how PIF4 phosphorylation affects its binding to BOPs. This shall help reveal the functional role of the BOP-mediated degradation and potentially address how the degradation is controlled by light.*

We agree that the role of the PIF4 phosphorylation is an important question. However, the situation is very different for PIF3 and PIF4. For PIF3 there are phospho-dead and phospho-mimic multisite mutants available that show the absolute requirement of PIF3 phosphorylation for binding to the E3 ligase LRB2 (Ni et al. 2014). However, no such mutants are available for PIF4. Bernardo-Garciá et al. (2014) have shown that mutations at three different putative phosphorylation sites somewhat modulates the stability of the PIF4 protein at dawn, probably associated with a brassinolide response. But it is not clear whether phosphorylation is an absolute requirement for degradation. Based on the western blot presented in this paper this mutant still presents clear light-induced phosphorylation as observed by the presence of a slower migrating band in the light. Because this PIF4 mutant is still phosphorylated in response to light we believe that it is not an efficient tool to address the role of PIF4 phosphorylation for binding to BOP and BOP-mediated degradation. Furthermore, our data clearly show that the situation is very different for the BOP-PIF4 interaction compared to the LRB2-PIF3 interaction since in our in vitro experiments phosphorylation is not required for binding (our *E. coli*-produced recombinant proteins are non-phosphorylated). This does not exclude that phosphorylation might be important to modulate the binding activity in vivo, but at least show a clear difference in the basal requirements for binding. We have expanded the Discussion in order to be clearer about this point.

Also, although the effect of BOPs on PIF4 accumulation is most visible in the light (Figure 3, Figure 5), our data is consistent with a role of BOPs that is not restricted to the light. Indeed, we observed a very low level of PIF4 ubiquitination in the dark, and we have included a new supplementary figure to show this (Figure 5—figure supplement 1). This ubiquitination was very much reduced in the *bop2* background (Figure 5—figure supplement 1). We therefore conclude that BOP2-mediated PIF4 degradation is not strictly dependent on light-induced PIF4 phosphorylation, however this regulatory mechanism has a strong influence on PIF4 abundance in the light. This situation is therefore profoundly different from LRB-mediated PIF3 degradation.

*2) Based on the phenotypes of bop1, bop2 and their double mutants, BOP1 and BOP2 may preferentially mediate PIF4 abundance under different environmental conditions. This manuscript at this stage does not present consistent biochemical data along with the genetic data. Almost all biochemical data are focused on BOP2, while genetic data usually include both bop1 and bop2. As the authors have by default included BOP1 and BOP2 in the E3 ubiquitin ligase complex, they should present evidence that BOP1 also affects PIF4 protein abundance under certain conditions.*

We initially characterized *bop* mutants in response to different light qualities and found out that BOP2 has a dominant role, which led us to concentrate on BOP2 (Figure 1, Figure 1—figure supplement 1). However, as the reviewers have pointed out, we have also found that BOP1 plays a role in other conditions, for example in 12/12 hour-grown plants and in high temperature. To be consistent we have now included data from 12/12 hour-grown plants to show that under these conditions PIF4 accumulation is also controlled by BOP1 (Figure 6—figure supplement 1).

*3) The authors should preferably provide co-IP data for the interaction between BOP1/2 with PIF4 and CUL3A using protein extracts from transgenic lines rather than Arabidopsis protoplasts.*

We agree that in general such data is preferable. However, such data have been very hard to generate because of the weak and specific expression of BOPs when expressed from their endogenous promoters. We have been able to generate data from transgenic plants for the BOP-CUL3 interaction using a CUL3 antibody (Enzo Life Sciences, Inc.) (Figure 4—figure supplement 1), but for the BOP-PIF4 interaction this would require protein extracts from transgenic plants simultaneously overexpressing tagged BOP and PIF4 proteins and unfortunately no such plants are yet available. However, we have confirmed the interaction between BOP2 and PIF4 with four independent methods (co-IP from *Arabidopsis* protoplasts, BiFC in Nicotiana leaves, in vitro interactions with purified recombinant proteins and yeast 2-hybrid assays). We have also demonstrated the interaction with the in vitroubiquitination assay. Since theses interaction data are consistent with both the genetic and other biochemical data we hope that this will be convincing enough.